# A Genome-Wide Association Study Identifies the Association between the 12q24 Locus and Black Tea Consumption in Japanese Populations

**DOI:** 10.3390/nu12103182

**Published:** 2020-10-18

**Authors:** Kyohei Furukawa, Maki Igarashi, Huijuan Jia, Shun Nogawa, Kaoru Kawafune, Tsuyoshi Hachiya, Shoko Takahashi, Kenji Saito, Hisanori Kato

**Affiliations:** 1Health Nutrition, Department of Applied Biological Chemistry, Graduate School of Agricultural and Life Sciences, The University of Tokyo, 1-1-1 Yayoi, Bunkyo-ku, Tokyo 113-8657, Japan; k-furukawa@g.ecc.u-tokyo.ac.jp (K.F.); igarashi.maki2020@mail.u-tokyo.ac.jp (M.I.); kkkj774@yahoo.co.jp (K.S.); 2Department of Molecular Endocrinology, National Research Institute for Child Health and Development, 2-10-1 Okura, Setagaya-ku, Tokyo 157-8535, Japan; 3Research and Development Department, Genequest Inc., 5-29-11 Siba, Minato-ku, Tokyo 108-0014, Japan; nogawa@genequest.jp (S.N.); kaoru.kawafune@genequest.jp (K.K.); hachiya@genome-analytics.co.jp (T.H.); takahashi@genequest.jp (S.T.); 4Department of Genomic Data Analysis Service, Genome Analytics Japan Inc., 15-1-3205 Toyoshima-cho, Shinjuku-ku, Tokyo 162-0067, Japan

**Keywords:** 12q24 locus, single nucleotide polymorphism, black tea, genome-wide association study, drinking frequency

## Abstract

Several genome-wide association studies (GWASs) have reported the association between genetic variants and the habitual consumption of foods and drinks; however, no association data are available regarding the consumption of black tea. The present study aimed to identify genetic variants associated with black tea consumption in 12,258 Japanese participants. Data on black tea consumption were collected by a self-administered questionnaire, and genotype data were obtained from a single nucleotide polymorphism array. In the discovery GWAS, two loci met suggestive significance (*p* < 1.0 × 10^−6^). Three genetic variants (rs2074356, rs144504271, and rs12231737) at 12q24 locus were also significantly associated with black tea consumption in the replication stage (*p* < 0.05) and during the meta-analysis (*p* < 5.0 × 10^−8^). The association of rs2074356 with black tea consumption was slightly attenuated by the additional adjustment for alcohol drinking frequency. In conclusion, genetic variants at the 12q24 locus were associated with black tea consumption in Japanese populations, and the association is at least partly mediated by alcohol drinking frequency.

## 1. Introduction

Black tea, which is derived from the *Camellia sinensis* plant, is one of the most widely consumed beverages in the world owing to its several beneficial compounds, including polyphenols, amino acids, vitamins, and minerals. Nutritional studies have demonstrated that theaflavin, a polyphenol, has positive effects against oxidative stress, endoplasmic reticulum (ER) stress, hypertension, aging, and diabetes [1,2,3,4], and although it remains controversial, a meta-analysis of epidemiological studies indicated that consuming three cups per day of black tea could prevent the onset of ischemic stroke [5].

Recently, genome-wide association studies (GWASs) have been conducted to investigate the association between genetic variants and the habitual intake of foods and drinks [6,7]. Previous studies identified the association of genetic variants with the intake frequency of fish [8], alcohol consumption [9,10,11], and coffee consumption [12,13,14]. Furthermore, dietary trait-associated genetic variants were linked to health outcomes in Japanese populations [7]. Thus, it is crucial to accumulate data from GWASs to identify the relationship between genetic variants and dietary habits. However, no data are available regarding black tea consumption-related genetic variants in individuals of other ethnicities, including Asian, European, and American. Therefore, this study performed a GWAS to search for genetic variants associated with the habitual consumption of black tea in 12,258 Japanese participants.

## 2. Materials and Methods

### 2.1. Study Design

The participants were enrolled from customers of a direct-to-consumer genetic testing service in Japan, “HealthData Lab,” provided by Yahoo! Japan Corporation (Tokyo, Japan) and Genequest Inc. (Tokyo, Japan). The inclusion criterion was age ≥ 18 years. Eligible participants were asked to complete internet-based questionnaires covering socio-demographic factors, medical history, and lifestyle habits at the time of enrollment. Among the 12,621 participants, one participant who opted out was excluded from the present study.

This study was approved by the Ethics Committee of Genequest Inc. (2015-0907-1) and was conducted according to the principles expressed in the Declaration of Helsinki. Written informed consent was obtained from all the participants for the general use of the questionnaires and their genotype data. An additional agreement was obtained with the opportunity to opt-out after informing the participants of this study’s aim.

### 2.2. Black Tea Consumption

The questionnaires included a question regarding black tea consumption, i.e., “How many cups of black tea do you drink?” with the following seven answers: (i) hardly any, (ii) less than or equal to two cups per week, (iii) three to four cups per week, (iv) five to six cups per week, (v) one to two cups per day, (vi) three to four cups per day, and (vii) more than or equal to five cups per day. These answers were converted into continuous variables (cups per day): (i) 0 cups, (ii) 0.14 (=1/7) cups, (iii) 0.5 (=3.5/7) cups, (iv) 0.79 (=5.5/7) cups, (v) 1.5 cups, (vi) 3.5 cups, and (vii) 5 cups. This continuous value was then used to define as the amount of habitual black tea consumption.

### 2.3. Adjustment Variables

Data on alcohol consumption, alcohol drinking frequency, habitual coffee consumption, sweet preference, and body-mass index (BMI) were used as adjustment variables. These data were obtained from the questionnaire, as described previously [8,12,15]. Briefly, alcohol drinking frequency was defined according to the question, “How frequently do you drink alcohol?” with the following seven answers: (i) barely ever, (ii) less than once a month, (iii) one to three times per month, (iv) one to two times per week, (v) three to four times per week, (vi) five to six times per week, and (vii) every day. Alcohol consumption was calculated by adding the amount of alcohol in grams per day obtained from beer, red wine, white wine, highballs/cocktails, rice wine, and distilled spirits.

Coffee consumption was defined based on the response to the question, “How many cups of coffee (instant or regular) do you drink?” and “How many cups of coffee (can, PET bottles, or paper pack) do you drink?” with a choice of seven answers: (i) hardly any, (ii) less than or equal to two cups per week, (iii) three to four cups per week, (iv) five to six cups per week, (v) one to two cups per day, (vi) three to four cups per day, and (vii) more than or equal to five cups per day. The sum of these values was used to define as the amount of habitual coffee consumption [12].

Sweet taste preference was determined using the question, “Please tell us about your taste for sweet food,” with the following five-point scale: (i) 1, dislike a lot, (ii) 2, dislike, (iii) 3, neither like nor dislike, (iv) 4, like, and (v) 5, like a lot.

### 2.4. DNA Sampling, Genotype, Quality Control, and Genotype Imputation

The collection, stabilization, and transportation of saliva samples were performed using an Oragene DNA Collection Kit (DNA Genotek Inc., Ottawa, Ontario, Canada). Genotype analysis was performed using The Illumina HumanCore-12 + Custom BeadChip (Illumina, San Diego, CA, USA), which contains 302,073 markers, and the HumanCore-24 + Custom BeadChip, which contains 309,725 markers. We used a total of 296,675 markers in the present study.

After excluding 17 subjects who lived outside of Japan, 12,603 participants remained. They were divided into two groups: those from the Hondo region (except for Okinawa region) and those from the Okinawa region, and they were referred to as the discovery and replication cohorts, respectively. Then, we applied the quality control and association analysis procedure separately for each cohort.

In the quality control analysis, we filtered out single nucleotide polymorphism (SNP) markers with low call rates (<0.95), low Hardy–Weinberg equilibrium exact test *p*-values (<1 × 10^−6^), or low minor allele frequencies (<0.01). We also excluded subjects who had inconsistent sex information between the genotype and questionnaire, who had an estimated non-Japanese ancestry, or who had a low call rate (<0.95) [16,17]. Furthermore, we excluded the close relationship pairs determined using the identity-by-descent method (PI_HAT > 0.1875) according to previous studies [8,13,18,19]. Quality control analyses were carried out using PLINK [20,21] (version 1.90b3.42) and Eigensoft [16] (version 6.1.3).

The imputation of the genotype was performed using the 1000 Genome Phase 3 (version 5) reference panel [22]. Pre-phasing was carried out using EAGLE2 (version 2.4) [23], and genotype imputation was performed using Minimac3 (version 2.0.1) software [24]. We excluded variants with a low imputation quality (R^2^ < 0.8) and a low minor allele frequency (<1%) from further analysis. Finally, we used the dosage data for the 5,256,047 variants for the GWAS in the discovery phase.

### 2.5. Genome-Wide Association and Meta-Analysis

The association between genotype dosage and the amount of black tea consumption was examined using a linear regression model under the assumption of additive genetic effects. For the discovery GWAS, the association was tested with an adjustment for age, sex, and five principal components calculated using PLINK software. Variants that met suggestive significance (*p* < 1.0 × 10^−6^) in the discovery phase were further investigated in the replication stage to determine their associations with habitual black tea consumption. Thereafter, we combined the statistical data from both stages using a fixed-effects model and the inverse-variance weighting method with METAL software (version 2011-03-25) [25]. Variants achieving genome-wide significance (*p* < 5.0 × 10^−8^) in the meta-analysis were considered to be associated with habitual black tea consumption.

### 2.6. Confounding Factor Adjustment and Subgroup Analysis

Multivariate linear regression analysis was carried out to test the associations between SNPs and black tea consumption with an additional adjustment for alcohol frequency, alcohol consumption, coffee consumption, sweet preference, and BMI. The subgroup analysis was carried out according to sex and age, and *P* < 0.05 was considered statistically significant in the subgroup analysis. These analyses were conducted based on the discovery populations.

## 3. Results

### 3.1. Research Flow and Characteristics of the Study Participants

Since the population of the Okinawa region forms a genetic cluster different from that of the other regions (Hondo region) in Japan [17], we investigated the effects of common variants on black tea consumption in 12,140 individuals for the discovery GWAS (Hondo region) and 118 individuals for the replication stage (Okinawa region), as illustrated in Figure 1. Thereafter, a meta-analysis from both the discovery and replication populations was carried out to identify black tea consumption-associated loci.

The characteristics of the Japanese participants from the discovery and replication stages are shown in Table 1. The mean age of the discovery and replication cohorts was 50.3 ± 13.2 years and 49.0 ± 12.3 years, respectively, and 46.8% and 45.8% of cohorts were female, respectively. The mean amount of black tea consumption per day was 0.20 ± 0.51 cups for the discovery population and 0.15 ± 0.41 cups for the replication population.

### 3.2. Discovery GWAS

The quantile-quantile plot indicated that the inflation factor (λ) was 1.005 and the 95% confidence interval was 1.003–1.007 in the discovery GWAS (Figure 2), indicating that the population structure was well-adjusted.

As illustrated in Figure 3, the discovery GWAS found that the 12q15 and 12q24 loci met suggestive significance (*p* < 1 × 10^−6^). Eleven SNPs at 12q24 locus and one SNP at 12q15 locus were detected as candidates for black tea consumption-associated variants (Table 2). The variants were positively associated with black tea consumption in the discovery stage.

### 3.3. Replication Stage and Meta-Analysis

The candidate variants found in the discovery GWAS were further examined in the replication stage. The eleven SNPs at 12q24 locus were associated with habitual black tea consumption (*p* < 0.05), whereas no significant association was observed in the case of rs1981764 (*p* = 0.81). A meta-analysis of both populations showed that three variants (rs2074356, rs144504271, and rs12231737) met genome-wide significance (*p* < 5 × 10^−8^), but other variants did not. With regard to the top SNP, rs2074356, the frequency of the T allele was 24.3%, and the effect size was estimated to be 0.042 (SE = 0.008) cups/day per allele in our meta-analysis. Significant heterogeneity in the genetic effects of rs2074356 was observed in the present study (I^2^ = 76.1 and *p*-value for heterogeneity = 0.04).

### 3.4. Adjustment for Potential Confounding Factors

We focused on the top SNP, rs2074356, for further analysis. The 12q24 locus is a strong long-range linkage disequilibrium (LD) region [26]. Previous GWASs in Asian populations showed the association of the 12q24 locus with dietary behaviors, preferences, and health outcomes [8,12,27,28]. Thus, the association of rs2074356 with black tea consumption might be confounded by the above-mentioned variables. Accordingly, we performed a multivariate linear regression analysis with further adjustment for dietary behaviors, preferences, and BMI. As shown in Table 3, the association between the 12q24 locus and black tea consumption was slightly attenuated by adding drinking frequency as an adjustment variable (*p*-value was changed from 1.2 × 10^−7^ to 3.8 × 10^−5^), although the association was still significant after the additional adjustment. In contrast, the addition of other variables hardly attenuated the association between the 12q24 locus and black tea consumption.

### 3.5. Subgroup Analysis According to Sex and Age

Subgroup analysis was conducted to examine whether the genetic effect of the variants at the 12q24 locus differed according to sex or age. In the subgroup analysis based on age, the median age of 51 years was used as the cutoff to categorize the individuals into the younger and older subgroups.

The results showed that the effect size of the rs2074356 variant was 0.028 (SE = 0.009) and 0.054 (SE = 0.013) cups per day in males and females, respectively. The genetic effect of the variant was slightly higher in females than in males, but the difference did not reach significance (*p* for interaction = 0.13; Figure 4a). No significant change was observed in the effect size between the younger subgroup (0.034, SE = 0.011) and the older subgroup (0.046, SE = 0.010) (*p* for interaction = 0.26; Figure 4b).

## 4. Discussion

The present study indicated an association of the 12q24 locus with black tea consumption in Japanese populations. Given that the 12q24 locus is specific to East Asian populations, this association cannot be found in studies of non-East Asian populations.

Regarding caffeine beverages, previous GWASs mainly investigated participants’ coffee consumption and identified several genetic variants in various populations [12,13,14]. Several studies also identified the variants related to caffeine intake, which was calculated by summing the consumption of caffeinated coffee, tea, soft drinks, and chocolate in individuals of European descent in populations from the United States and Costa Rica [29,30], whereas no GWAS data were available on the variants that are specific to black tea consumption. To the best of our knowledge, the present study is the first GWAS of habitual black tea consumption.

Furthermore, the 12q24 locus affects several dietary habits in East Asian populations [7,8,12,13,31]. Our analysis indicated that although the association between the 12q24 locus and black tea consumption was slightly attenuated by the further adjustment for drinking frequency, the association was still significant after the adjustment. Previous studies have indicated that the association of the 12q24 variant with fish intake frequency [8] (*p*-value was changed from 4.3 × 10^−11^ to 2.5 × 10^−4^) and sweet preference [15] (*p*-value from 1.2 × 10^−69^ to 6.5 × 10^−10^) was largely attenuated by adding drinking frequency as an adjustment variable. Moreover, the association of the 12q24 locus with black tea consumption was not evidently attenuated by additional adjustment for alcohol consumption. Thus, the association of the 12q24 locus with black tea consumption may be partly confounded by drinking frequency; regardless, the 12q24 locus certainly had a direct effect on black tea consumption.

In our meta-analysis, the rs2074356 SNP at the 12q24 locus had a beta value of 0.042, which indicated that Japanese participants’ black tea intake increased by 0.042 cups per day, 1.26 cups per month, and more than 15 cups per year for each copy of the minor allele. Considering that black tea contains several beneficial nutrients and other compounds, it is conceivable that the variants have health-promoting effects in East Asian populations. Particularly, caffeine is widely known to have health benefits related to type 2 diabetes mellitus and glucose tolerance [31]. Previous studies indicated that rs2074356 SNP was associated with coffee consumption in Japanese [13] and Korean populations [32], suggesting that the SNP could be associated with caffeine intake in East Asian populations. Theaflavin, a polyphenol in black tea, was also reported to suppress oxidative stress and hyperglycemia in a streptozotocin-induced diabetic model [4] and ER stress and hypertension in an angiotensin II-induced hypertension model [2]. Furthermore, a recent GWAS in Japanese populations indicated a strong association of the 12q24 locus with liver disorder-related parameters (alanine transaminase and aspartate transaminase) and blood pressure [7]. With regard to black tea-associated SNPs, HECTD4 is a ubiquitin ligase, and TRAFD1 relates to the toll-like receptor 4 signaling pathway [33], whereas there are limited data on the association between health outcomes and the three SNPs. Therefore, it is still important to study the complicated relationship between genetic background at 12q24 locus, dietary and drinking habits, and health outcomes.

Recently, GWASs have demonstrated genetic variants associated with dietary habits in several populations, and the results of these studies could be utilized for personalized nutritional recommendations [34]. In a Japanese population, a previous GWAS indicated that the 12q24 locus was associated with several dietary habits and health outcomes [7]. Thus, the genotype of the 12q24 locus may be a useful factor for personalized nutritional recommendations. In future studies, it is pertinent to accumulate GWAS data to develop novel personalized nutritional recommendations.

This study had the following two limitations. First, the 12q24 locus is specific to East Asian populations; however, our study investigated only Japanese populations. Second, we could not deny the possibility that the association between the 12q24 locus and black tea consumption may be confounded by unmeasured factors, including health outcomes and dietary factors. Therefore, further research with several populations and unmeasured factors are still required to determine the impacts of the genetic variants on black tea consumption and its related health outcomes.

## 5. Conclusions

We found that genetic variants at the 12q24 locus were associated with black tea consumption in Japanese populations. This finding not only provided fundamental information with regard to the association between genetic variants and dietary habits but also contributed to the development of novel strategies for personalized nutritional recommendations.

## Figures and Tables

**Figure 1 nutrients-12-03182-f001:**
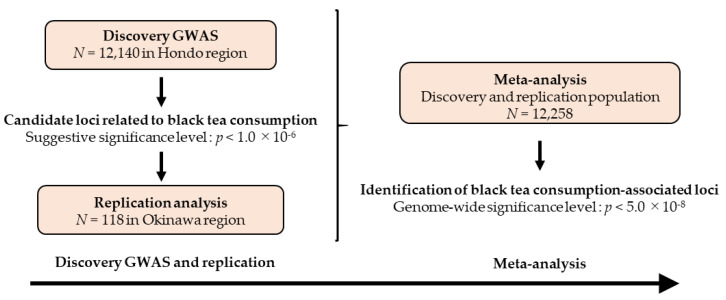
Research flow of the present study. GWAS: Genome-wide association study.

**Figure 2 nutrients-12-03182-f002:**
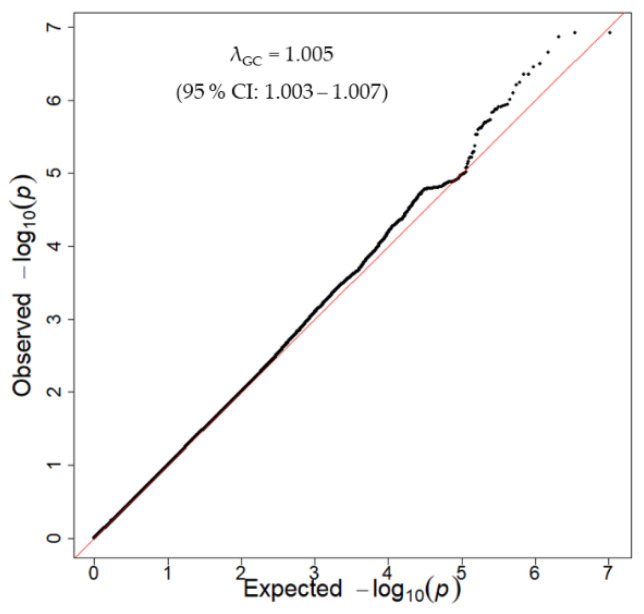
Quantile-quantile plot for the discovery GWAS. The *x*-axis and *y*-axis show the theoretical –log_10_
*p*-value and the observed –log_10_
*p*-value, respectively. The red line represents y = x. CI: confidence interval, GWAS: genome-wide association study.

**Figure 3 nutrients-12-03182-f003:**
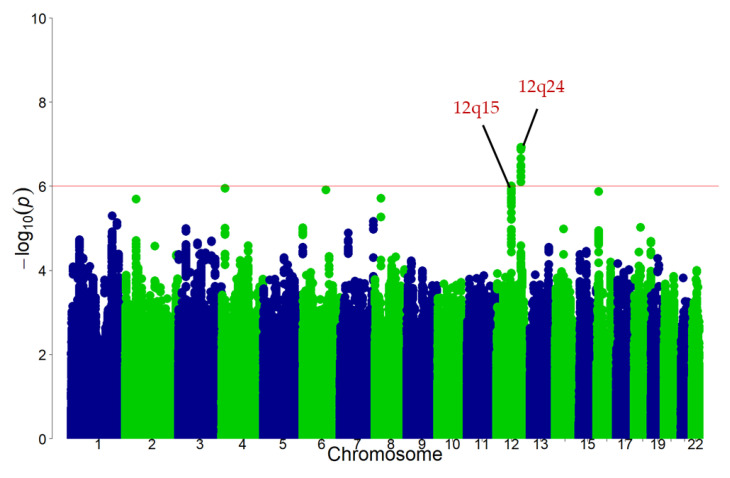
Manhattan plot for the discovery GWAS of habitual black tea consumption in the Japanese population. Single nucleotide polymorphism (SNPs) are ordered by chromosome and position in the *x*-axis. The *y*-axis shows the negative logarithm of the association of each SNP with black tea consumption. The red line represents *p* < 1.0 × 10^−6^. GWAS: genome-wide association study.

**Figure 4 nutrients-12-03182-f004:**
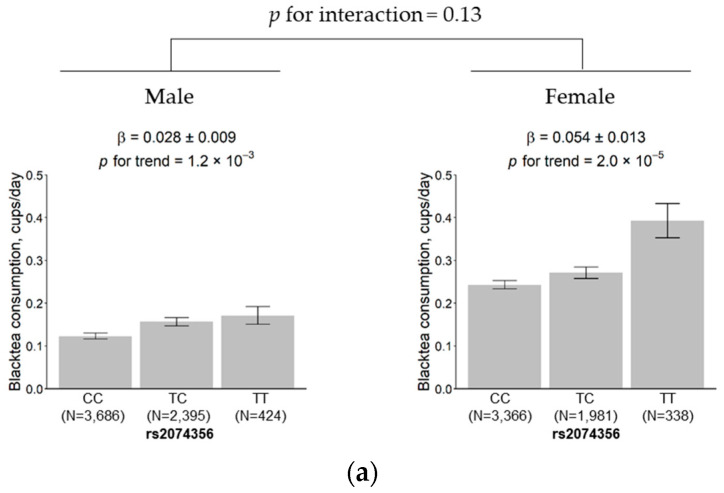
Effects of rs2074356 on habitual black tea consumption stratified by sex (**a**) and age (**b**). The black tea consumption is marked for each allele, CC, TC, and TT, for rs2074356. Error bars and β indicate the standard deviation and the estimated regression coefficient, respectively.

**Table 1 nutrients-12-03182-t001:** Participant characteristics.

Parameters	Discovery	Replication
*N*	12,140	118
Female (%)	46.8	45.8
Age, years (mean ± SD)	50.3 ± 13.2	49.0 ± 12.3
Black tea consumption, cups/day (mean ± SD)	0.20 ± 0.51	0.15 ± 0.41
Drinking frequency, times/week (mean ± SD)	2.21 ± 2.67	2.63 ± 2.76
Alcohol consumption, g/day (means ± SD)	7.10 ± 11.94	9.87 ± 15.02
Coffee consumption, cups/day (mean ± SD)	1.70 ± 1.50	1.64 ± 1.28
Sweet taste preference, (mean ± SD)	3.74 ± 0.90	3.72 ± 0.89
BMI, kg/m^2^ (mean ± SD)	23.1 ± 3.7	23.9 ± 4.0

BMI: body mass index, SD: standard deviation.

**Table 2 nutrients-12-03182-t002:** Variants associated with habitual black tea consumption.

SNP	Chr	Position	Gene	EA	NEA	Population	EAF	Beta	SE (Beta)	*p* _Association_
rs2074356	12	112645401	*HECTD4*	T	C	Discovery	0.244	0.040	0.008	1.2 × 10^−7^
						Replication	0.174	0.184	0.070	0.01
						Meta-analysis	0.243	0.042	0.008	2.4 × 10^−8^
rs144504271	12	112627350	*HECTD4*	A	G	Discovery	0.264	0.041	0.008	1.2 × 10^−7^
						Replication	0.190	0.183	0.072	0.01
						Meta-analysis	0.263	0.042	0.008	3.1 × 10^−8^
rs12231737	12	112574616	*TRAFD1*	T	C	Discovery	0.266	0.041	0.008	1.4 × 10^−7^
						Replication	0.192	0.187	0.072	0.01
						Meta-analysis	0.265	0.043	0.008	4.3 × 10^−8^
rs116873087	12	112511913	*NAA25*	C	G	Discovery	0.263	0.041	0.008	2.2 × 10^−7^
						Replication	0.189	0.186	0.074	0.01
						Meta-analysis	0.262	0.043	0.008	5.8 × 10^−8^
rs11066132	12	112468206	*NAA25*	T	C	Discovery	0.261	0.040	0.008	3.2 × 10^−7^
						Replication	0.188	0.188	0.074	0.01
						Meta-analysis	0.260	0.042	0.008	9.6 × 10^−8^
rs78069066	12	112337924	*MAPKAPK5* *TMEM116*	A	G	Discovery	0.267	0.039	0.008	3.5 × 10^−7^
						Replication	0.189	0.177	0.072	0.01
						Meta-analysis	0.266	0.040	0.008	1.1 × 10^−7^
rs4646776	12	112230019	*ALDH2*	C	G	Discovery	0.264	0.037	0.007	4.4 × 10^−7^
						Replication	0.187	0.171	0.070	0.02
						Meta-analysis	0.263	0.039	0.007	1.4 × 10^−7^
rs671	12	112241766	*ALDH2*	A	G	Discovery	0.264	0.037	0.007	4.5 × 10^−7^
						Replication	0.186	0.171	0.070	0.02
						Meta-analysis	0.263	0.039	0.007	1.5 × 10^−7^
rs11066001	12	112119171	*BRAP*	C	T	Discovery	0.262	0.038	0.008	6.2 × 10^−7^
						Replication	0.185	0.175	0.072	0.02
						Meta-analysis	0.261	0.039	0.008	1.6 × 10^−7^
rs11066015	12	112168009	*ACAD10*	A	G	Discovery	0.263	0.037	0.007	5.7 × 10^−7^
						Replication	0.187	0.171	0.070	0.02
						Meta-analysis	0.262	0.038	0.007	1.8 × 10^−7^
rs3782886	12	112110489	*BRAP*	C	T	Discovery	0.264	0.037	0.008	7.9 × 10^−7^
						Replication	0.186	0.175	0.072	0.02
						Meta-analysis	0.263	0.039	0.008	2.6 × 10^−7^
rs1981764	12	68217659	*DYRK2– IFNG*	G	A	Discovery	0.123	0.049	0.010	9.9 × 10^−7^
						Replication	0.119	−0.021	0.087	0.81
						Meta-analysis	0.127	0.048	0.010	1.4 × 10^−6^

Chromosomal positions are based on the human genome assembly version GRCh37/hg19. SNP: single nucleotide polymorphism, Chr: chromosome, EA: effect allele, NEA: non-effect allele, EAF: effect allele frequency, SE: standard error.

**Table 3 nutrients-12-03182-t003:** Additional adjustment for confounding factors.

Adjustment Variables	Beta	SE (Beta)	p _Association_
Age, Sex, Population structure (5 PCs)	0.040	0.008	1.2 × 10^−7^
Age, Sex, Population structure (5 PCs), Drinking frequency	0.033	0.008	3.8 × 10^−5^
Age, Sex, Population structure (5 PCs), Alcohol consumption	0.037	0.008	3.1 × 10^−6^
Age, Sex, Population structure (5 PCs), Coffee consumption	0.045	0.008	9.9 × 10^−9^
Age, Sex, Population structure (5 PCs), Sweet preference	0.039	0.008	3.0 × 10^−7^
Age, Sex, Population structure (5 PCs), BMI	0.040	0.008	9.9 × 10^−8^

SE: standard error, BMI: body mass index, PC: principal component.

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
