# Peer review of "A Genome-Wide Association Study Identifies the Association between the 12q24 Locus and Black Tea Consumption in Japanese Populations"

_nutrients, 2020, doi:10.3390/nu12103182_

Round 1

Reviewer 1 Report

In this study, the authors performed GWAS analysis in order to investigate the association of genetic variants with black tea consumption in Japanese populations. An important point if we assume that a higher consumption could prevent the onset of several clinical condition such as hypertension, aging, diabetes, ischemic stroke etc. The paper is well written and the methods used are appropriate. However, the results are not totally convincing.

  • The authors found a strong association with different variants in the 12q24 locus however, they pay attention only to one, the rs2074356 G>A in HECTD4 gene. Are the other variants showed in figure 3 all located in the same gene? Why aren't they listed?
  • The 12q24 locus include more genes? Since recent GWAS experiments in the same population indicated a strong association of the same locus with liver disorder-related parameters, blood pressure and diabet could be important to show the implicated genes.
  • Is the choice of the variant due only to the higher significance of association or to the fact that the same variant has already been described to be associated with a greater consumption of coffee? PMID: 32722627.
  • Moreover, the right nomenclature of the variant is C>T while the authors use the complementary sequence G>A.

Then, only if all variants and genes at 12q24 locus will be listed, the variant will be well named and a sentence regarding the recent publication about coffee and its-related health outcomes will be added in the discussion.

Author Response

Reviewer 1

We appreciate your kind and valuable comments on our manuscript and have answered all the comments point-by-point as follows. Please see the yellow highlighted parts.

  • The authors found a strong association with different variants in the 12q24 locus however, they pay attention only to one, the rs2074356 G>A in HECTD4 gene.Are the other variants showed in figure 3 all located in the same gene? Why aren't they listed?

Thank you for your suggestion. We have added ten SNPs at 12q24 locus that met suggestive significance in the discovery GWAS (Table 2). Of these, three SNPs were also significantly associated with black tea consumption in the replication stage and meta-analysis. Two of the three SNPs were located in the HECTD4 gene and the remaining SNP was located in TRAFD1.

  • The 12q24 locus include more genes? Since recent GWAS experiments in the same population indicated a strong association of the same locus with liver disorder-related parameters, blood pressure and diabet could be important to show the implicated genes.

The 12q24 locus is a strong long-range linkage disequilibrium region and has been reported to be associated with several eating and drinking behaviors and health outcomes in the East-Asian population. However, there is limited information regarding the relationship of the HECTD4 and TRAFD1 genes, which met genome-wide significance, with health outcomes. Therefore, it is hard to discuss the relationship between health outcomes and black tea consumption-related SNPs, but a sentence was added in the discussion section (Page 9, Lines 228-235).

Original sentence

Recent a GWAS in Japanese populations indicated a strong association of the 12q24 locus with liver disorder-related parameters (alanine transaminase and aspartate transaminase) and blood pressure [7]. These pleiotropic effects of the 12q24 locus suggest a complicated relationship between genetic background, dietary and drinking habits, and health outcomes.

Revised sentence

Furthermore, recent a GWAS in Japanese populations indicated a strong association of the 12q24 locus with liver disorder-related parameters (alanine transaminase and aspartate transaminase) and blood pressure [7]. With regard to black tea-associated SNPs, HECTD4 is an ubiquitin ligase and TRAFD1 relates to toll-like receptor 4 signaling pathway [33], while there are limited data on the association between health outcomes and the three SNPs. Therefore, it is still important to study the complicated relationship between genetic background at 12q24 locus, dietary and drinking habits, and health outcomes.

  • Is the choice of the variant due only to the higher significance of association or to the fact that the same variant has already been described to be associated with a greater consumption of coffee? PMID: 32722627.

Thank you for providing this article. We added the sentence in the discussion section (Page 9, Line 222-226).

Original sentence

caffeine is widely known to have health benefits related to type 2 diabetes mellitus and glucose tolerance [31]

Revised sentence

Particularly, caffeine is widely known to have health benefits related to type 2 diabetes mellitus and glucose tolerance [31]. Previous studies indicated that rs2074356 SNP was associated with coffee consumption in Japanese [13] and Korean populations [32], suggesting that the SNP could be associated with caffeine intake in East Asian populations.

  • Moreover, the right nomenclature of the variant is C>T while the authors use the complementary sequence G>A.

Thank you for pointing this out. We modified rs20754256 information from G>A to C>T.

Reviewer 2 Report

Dear Respected Authors.

Thank you for you contribution to the journal. The manuscript has been written very well without major flaws.

Abstract, title and references

  • Is the aim clear? Yes ● Is it clear what the study found and how they did it? Yes ● Is the title informative and relevant? Yes ● Are the references: ● Relevant? Yes ● Recent? Yes ● Referenced correctly? Yes ● Are appropriate key studies included? Yes

Introduction/ background

  • Is it clear what is already known about this topic? Yes ● Is the research question clearly outlined? Yes ● Is the research question justified given what is already known about the topic? Yes

Methods

  • Is the process of subject selection clear? Yes ● Are the variables defined and measured appropriately? Yes ● Are the study methods valid and reliable? To some Extent ● Is there enough detail in order to replicate the study? Yes

Results

  • Is the data presented in an appropriate way? Yes ● Tables and figures relevant and clearly presented? Yes ● Appropriate units, rounding, and number of decimals? Yes ● Titles, columns, and rows labelled correctly and clearly? Yes ● Categories grouped appropriately? Yes ● Does the text in the results add to the data or is it repetitive? repetitive ● Are you clear about what is a statistically significant result? Yes ● Are you clear about what is a practically meaningful result? Yes

Discussion and Conclusions

  • Are the results discussed from multiple angles and placed into context without being over interpreted? Yes ● Do the conclusions answer the aims of the study? To some extent ● Are the conclusions supported by references or results? Yes ● Are the limitations of the study fatal or are they opportunities to inform future research? Needs Future Research.

Author Response

Thank you for your positive feedback.